# Major Depressive Disorder and Gut Microbiota: Role of Physical Exercise

**DOI:** 10.3390/ijms242316870

**Published:** 2023-11-28

**Authors:** Pedro Borges de Souza, Laura de Araujo Borba, Louise Castro de Jesus, Ana Paula Valverde, Joana Gil-Mohapel, Ana Lúcia S. Rodrigues

**Affiliations:** 1Center of Biological Sciences, Department of Biochemistry, Universidade Federal de Santa Catarina, Florianópolis 88037-000, SC, Brazil; pebsoza@gmail.com (P.B.d.S.); lauraborba28@gmail.com (L.d.A.B.); louisecastroj@gmail.com (L.C.d.J.); valverde.laned@gmail.com (A.P.V.); 2Island Medical Program, Faculty of Medicine, University of British Columbia, Victoria, BC V8P 5C2, Canada; 3Division of Medical Sciences, University of Victoria, Victoria, BC V8P 5C2, Canada

**Keywords:** gut microbiota, major depressive disorder, physical exercise

## Abstract

Major depressive disorder (MDD) has a high prevalence and is a major contributor to the global burden of disease. This psychiatric disorder results from a complex interaction between environmental and genetic factors. In recent years, the role of the gut microbiota in brain health has received particular attention, and compelling evidence has shown that patients suffering from depression have gut dysbiosis. Several studies have reported that gut dysbiosis-induced inflammation may cause and/or contribute to the development of depression through dysregulation of the gut–brain axis. Indeed, as a consequence of gut dysbiosis, neuroinflammatory alterations caused by microglial activation together with impairments in neuroplasticity may contribute to the development of depressive symptoms. The modulation of the gut microbiota has been recognized as a potential therapeutic strategy for the management of MMD. In this regard, physical exercise has been shown to positively change microbiota composition and diversity, and this can underlie, at least in part, its antidepressant effects. Given this, the present review will explore the relationship between physical exercise, gut microbiota and depression, with an emphasis on the potential of physical exercise as a non-invasive strategy for modulating the gut microbiota and, through this, regulating the gut–brain axis and alleviating MDD-related symptoms.

## 1. Introduction

Major depressive disorder (MDD) is a multifactorial psychiatric disorder characterized by at least one depressive episode for a minimum of 2 weeks. The essential feature of a depressive episode is the presence of either a depressed mood or anhedonia accompanied by other neurocognitive and neurovegetative symptoms, such as diminished ability to think or concentrate and sleep disturbances [1]. Approximately 4–5% of the world population is affected by MDD [1]. Individuals with MDD are at a higher risk of suicidality [2], developing physical comorbidities (e.g., cardiovascular, stroke, diabetes, obesity) [3], and negative outcomes in other social aspects of life, such as education, employment and personal relationships [4].

MDD is a complex disorder whose mechanisms are not completely established [1]. Animal models have been an important strategy for investigating the molecular pathways implicated in the pathophysiology of MDD. These pathways include, but are not limited to, genetic and epigenetic alterations, neuroendocrine and immune disturbances, neuroplasticity impairments, mainly with regard to neurogenesis, and alterations in the microbiota–gut–brain axis [5]. An increased understanding of these biological processes may assist in developing new strategies for managing MDD.

The response to stressful stimuli, either psychological or physical, is mediated by the hypothalamic–pituitary–adrenal (HPA) axis [6]. The termination of the stress response is based on a negative feedback mechanism and is mediated by the connection between the hippocampus and the paraventricular nucleus of the hypothalamus [7]. A healthy response to stress requires both a rapid and vigorous response followed by the termination of such response [7]. However, a dysfunction of the HPA axis is present in individuals with MDD [1]. In addition, HPA axis activation may alter the composition of the gut microbiota and increase gastrointestinal permeability [8].

Individuals with MDD present higher serum levels of inflammatory markers [9]. Administration of lipopolysaccharide (LPS) in rodents has been used to mimic the inflammatory response and depressive symptoms observed in individuals with MDD [10]. LPS inflammatory effects extend from the periphery to the brain. Peripherally, pro-inflammatory cytokine production increases, and cytokines can subsequently reach the brain either via macrophage-like cell-mediated signaling or by crossing the blood–brain barrier (BBB), resulting in microglial activation [10]. Microglia are responsible for scavenging damaged brain tissue and pathogenic agents. At rest, they present ramifications that help their function as sentinels. In response to threats, microglia switch to their active phenotype. This results in the activation of the NOD-like receptor pyrin domain-containing-3 (NLRP3) inflammasome and the consequent release of the inflammatory cytokines interleukin (IL)-1β and IL-18, which contributes to neuroinflammation as shown in Figure 1 [11]. Accordingly, image studies have identified increased neuroinflammation in the brain of MDD patients as compared with controls [12,13].

Another key inflammatory mechanism implicated in LPS-induced depressive-like behavior is the tryptophan (Trp)–kynurenine metabolism [14]. Trp metabolism can follow two directions: production of serotonin by tryptophan hydroxylase or production of kynurenine (KYN) by indoleamine 2,3-dioxygenase (IDO) and tryptophan 2,3-dioxygenase (TDO) enzymes [15]. The conversion of Trp into KYN is activated under pro-inflammatory conditions [14]. KYN metabolites can be grouped into two pathways according to their effects: the excitotoxic pathway and the neuroprotective pathway [11]. KYN excitotoxic metabolites are induced in inflammatory states and include 3-hydroxy-kynurenine (3-HK), 3-hydroxy-anthralinic acid, quinolinic acid (QUIN) (a glutamate receptor agonist) and nicotinamide adenine dinucleotide (NAD+). Neuroprotective metabolites include kynurenic acid (KYNA), with opposing effects to QUIN by its antagonistic action on the N-methyl-D-aspartate (NMDA) glutamatergic receptor and reduction in extracellular glutamate release [11]. In MDD, there is an imbalance in the Trp-KYN pathway, favoring the production of neurotoxic over neuroprotective metabolites as illustrated in Figure 1 [16,17]. 

The management of MDD consists of a wide range of pharmacological and non-pharmacological interventions. For mild MDD, lifestyle interventions such as physical exercise, alone or in combination, may provide significant antidepressant effects [1]. As such, this narrative review will discuss the role of physical exercise in the management of MDD, focusing on the possible mechanisms underlying its effects, particularly those related to the modulation of the gut microbiota.

## 2. Gut Microbiota and MDD

The gut is the biggest digestive, immune and endocrine organ of the human body and can sometimes be referred to as the second brain [18]. The gut is a microbial organ, and it is estimated that our gut harbors about 10^14^ microorganisms [19]. The first signature microorganisms are acquired at birth and develop over the course of the first few years of life. Although it is easily modulated by host genetic and environmental factors, such as diet, stress and exposure to other microorganisms or antibiotics, the microbiota adapts to the host, carrying out several important metabolic and biochemical processes. Thus, gut microorganisms can directly influence human health [20].

The bidirectional connections between gut microorganisms and the brain through various biological systems are called the microbiota–gut–brain axis. These connections are fundamental for the maintenance of gastrointestinal, central nervous system (CNS) and microbial homeostasis and occur through direct and indirect communication via the autonomic nervous system, enteric nervous system, neuroendocrine system and immune system and signaling through microbial-derived metabolites and products, chemical transmitters, and neuronal pathways [21]. 

This close communication between the intestinal microbiota and the CNS is sensitive to several factors, mainly environmental ones, including diet, antibiotic use, stress and infections [22]. Disturbances in the homeostasis of these systems have already been related to the pathophysiology of MDD [23].

Clinical evidence has shown that MDD individuals have an altered microbiota composition when compared to healthy controls, including an increased Bacteroidetes/Firmicutes ratio, which is considered a dysbiosis signature [24]. Of note, a meta-analysis study has demonstrated a decreased abundance of the bacterial families Veillonellaceae, Prevotellaceae and Sutterellaceae, as well as the genera *Coprococcus*, *Faecalibacterium*, *Ruminococcus*, *Bifidobacterium* and *Escherichia*, and an increased abundance of the family Actinomycetaceae and the genus *Paraprevotella* in MDD patients compared to controls [23]. In addition, preclinical studies have shown that probiotics could improve depression-like phenotypes. For example, Tian et al. [25] reported that ingestion of the *Bifidobacterium longum* subspecies *infantis* strain CCFM687 improved stress-induced depressive-like behavior, increased BDNF levels and the abundance of butyrate-producing bacteria and modulated the HPA axis in mice. Also, treatment with *Akkermansia muciniphila* improved chronic-stress-induced depressive-like behavior, modulated corticosterone, dopamine, and BDNF levels and regulated gut microbiota and metabolites in mice [26].

The most direct communication between the gut and brain is through the vagus nerve (cranial nerve X) [27]. This peripheral nervous system (PNS) nerve has an important role in sensory and parasympathetic regulation of gut physiology regulating motility, digestion and tonic secretion of gastric mucus via the neurotransmitter acetylcholine (ACh) [28,29,30]. Vagal neurons synapse with intestinal enteroendocrine cells (EECs), such as neuropod cells that can release neurotransmitters including glutamate and serotonin, glucagon-like peptide-1 (GLP-1) and peptide YY (PYY). These transmit sensory stimuli from the gut to the brain in milliseconds [20,31]. Several studies have suggested that abnormal composition of the gut microbiota is related to depressive and anhedonia-like phenotypes, and this is, at least in part, mediated by vagus nerve communication [32,33]. In fact, intraperitoneal injection of LPS in rats induced a depression-like phenotype that was abolished by subdiaphragmatic vagotomy [34]. Also, depression-like phenotypes, altered microbiota composition, systemic inflammation and downregulation of synaptic proteins in the medial prefrontal cortex were shown to be dependent on the subdiaphragmatic vagus nerve in mice exposed to LPS [35].

The gut microbiota is essential to the digestion of food, absorption of nutrients and production of metabolites like short-chain fatty acids (SCFAs), lipids, vitamins, bile acids, branched-chain amino acids, Trp and indole derivatives [36,37]. SCFAs are one of the most well-characterized metabolites produced by gut microbiota. They are saturated fatty acids with a carbon chain ranging from one to six atoms in length produced through the fermentation of dietary fiber in the colon, with acetate (C2), propionate (C3) and butyrate (C4) being the most common SCFAs [38]. After being absorbed by the colonocytes, SCFAs serve as an energy source, through the production of ATP in the mitochondria, being substrates for the synthesis of cholesterol and fatty acids. Moreover, SCFAs may improve gut and brain health and maintain the integrity of the intestinal and blood–brain barriers by regulating tight junction proteins, affecting mucus production and influencing gastrointestinal motility and appetite through the regulation of neuronal activity and intestinal hormones, such as GLP-1 and PYY. Also, SCFAs can protect from inflammation by modulating pro-inflammatory cytokines and chemokines, inhibiting nuclear factor kappa B (NF-kB) as well as histone deacetylases (HDACs) [39] and regulating microglial homeostasis in the CNS [40,41]. The main mechanism by which SCFAs seem to exert their effects is by activating G-protein-coupled receptors (GPRs) that are expressed in several cell types, including immune cells, adipocytes, skeletal and heart muscle cells, intestinal cells and brain cells [42,43,44]. More specifically, acetate, propionate and butyrate bind to GPR43 and GPR41 (also known as free fatty acid receptor 2 (FFAR2) and FFAR3, respectively), with butyrate also binding to GPR109A (also known as hydroxycarboxylic acid receptor 2 (HCAR2)) [45]. Among other functions, GPR43 and GPR41 are involved in the production and secretion of GLP-1 and PYY and exert anti-inflammatory and anti-tumorigenic activities [46,47]. 

Butyrate, the only known SCFA agonist for GPR109A, appears to be the principal agonist in the gastrointestinal tract [48], but this receptor can also be activated by niacin (also known as nicotinic acid and vitamin B3) and β-hydroxybutyrate [49,50]. Among other functions, GPR109A is involved in the reduction in chemokine and pro-inflammatory cytokine production, including IL-1β, IL-6 and tumor necrosis factor α (TNF-α) [51,52], as well as the inhibition of the NLRP3 inflammasome [53]. Furthermore, GPR109A can enhance the activity of adenosine monophosphate (AMP)-dependent kinase (AMPK) in microglia, resulting in the activation of sirtuin 1 (SIRT1), which inhibits NF-κB signaling via acetylation [54]. The nuclear factor (erythroid-derived) related factor-2 (Nrf2), which mediates antioxidant and anti-inflammatory signaling, has also been implicated in the pathways associated with the activation of this receptor [55]. Finally, GPR109A signaling has also been shown to be involved in the increase in several neurotrophic factors, including vascular endothelial growth factor (VEGF) and brain-derived neurotrophic factor (BDNF) [56,57]. All these pathways have been implicated in the pathophysiology of MDD, suggesting that butyrate production by the gut microbiota and activation of GPR109A may improve depressive symptoms.

SCFAs not only play an important role in CNS homeostasis by maintaining the integrity of the BBB but also have the ability to cross it, where they can regulate brain development, neuroplasticity, neurotransmitter synthesis, epigenetic factors and gene expression, as well as the immune response [21,40]. Indeed, sodium butyrate was shown to prevent microglia activation and depressive-like behaviors in mice [58], and in vitro studies have demonstrated its anti-inflammatory role through the decrease in LPS-induced microglial inflammation [59] in rat primary microglia cultures, hippocampal slices and co-cultures of rat cerebellar granule neurons, astrocytes, and microglial cells [60]. Furthermore, SCFAs have also been shown to reduce depressive-like behaviors by regulating HPA activity. In agreement, oral administration of acetate, propionate and butyrate was able to improve changes in intestinal permeability, HPA hyperactivity and anhedonia in mice submitted to repeated psychosocial stress [61].

As mentioned above, dysfunctions in neuroplasticity and neurogenesis have also been implicated in the pathophysiology of MDD. SCFAs have the capacity to modulate neurotrophins such as BDNF, nerve growth factor (NGF) and glial cell line-derived neurotrophic factor (GDNF) that regulate the growth, survival and differentiation of neurons and synapses, thereby impacting neuroplasticity [62,63]. In fact, subcutaneous administration of sodium butyrate was shown to induce cell proliferation, migration and differentiation in the hippocampal dentate gyrus in a rat model of ischemia [64]. Moreover, a combination of sodium butyrate and pyridoxine promoted cell proliferation and neurogenesis in mice [65]. In addition, systemic administration of sodium butyrate has been reported to induce histone hyperacetylation in the hippocampus and frontal cortex, upregulate BDNF transcript levels and elicit antidepressant-like effects [66].

Another important metabolite of the gut microbiota is lactate, which is produced by the fermentation of dietary fibers by lactic acid bacteria, such as the genera *Lactobacillus* and *Bifidobacterium*. Lactate can be further converted into different SCFAs by several bacterial species, including *Eubacterium hallii*, *Anaerostipes* spp. and *Veillonella* spp. [67,68]. Studies have suggested that lactate can be absorbed and cross the BBB and has an important role in CNS, being used as an energy substrate by neurons and contributing to synaptic plasticity [69]. Several lines of evidence have indicated that lactate abnormalities may be associated with MDD. Indeed, an increased urine lactate concentration was observed in severe MDD patients [70]. Moreover, lactate was able to improve stress-induced depressive-like behavior, and this effect was associated with changes in the expression of target genes involved in serotonin receptor trafficking, astrocyte functions, neurogenesis, nitric oxide synthesis and cAMP signaling [71]. Further studies have also supported the antidepressant effect of lactate. Karnib et al. [72] demonstrated that lactate was able to improve depressive-like behaviors and mediate resilience to stress by modulating hippocampal levels and activity of HDACs in mice exposed to social defeat stress. In addition, Carrard et al. [73] showed that L-lactate administration reversed corticosterone-induced depressive-like behavior and promoted the proliferation and survival of new hippocampal neurons in adult mice. Furthermore, pharmacological inhibition of adult hippocampal neurogenesis abolished this effect, demonstrating the role of neurogenesis in the antidepressant-like effect of lactate. In fact, lactate is involved in regulating neuronal plasticity-related genes, including those coding for BDNF, activity-regulated cytoskeleton-associated protein (ARC) and VEGF [74,75].

The gut microbiota can also regulate Trp metabolism and the KYN pathway in MDD, promoting a decrease in KYN and an increase in QUIN levels [76]. In the brain, the enzyme kynurenine aminotransferase (KAT) that converts KYN into KYNA is localized mainly in astrocytes, and microglia and macrophages are responsible mainly for the production of QUIN by kynureninase under inflammatory conditions [77]. Some microbial components present within the gut microbiota, such as LPS and lipoteichoic acids, can activate toll-like receptors (TLRs) and have been identified as key factors in initiating Trp metabolism through the KYN pathway [78]. Furthermore, SCFAs can modulate intestinal barrier integrity and systemic inflammation, which are in turn associated with an altered KYN pathway [79]. In addition, butyrate plays a role in reducing intestinal IDO expression by inhibiting HDAC and IFN-gamma-dependent phosphorylation of signal transducer and activator of transcription 1 (STAT1) and, subsequently, the STAT1-driven transcriptional activity of IDO [80]. 

The modulation of the KYN pathway by the gut microbiota may be implicated in abnormal KYN levels found in MDD patients. In fact, fecal microbiota transplantation (FMT) from MDD patients to antibiotic-treated rats induced anhedonia and anxiety-like behaviors along with decreased gut microbiota richness and diversity and an elevated KYN/Trp ratio [81]. In addition, a reduction in the Firmicutes phylum and a reduction in SCFA synthesis, which has been associated with increased inflammation and the diversion of KYN metabolism to the neurotoxic pathway with the consequent production of QUIN, have been observed in MDD patients [82,83].

## 3. Physical Exercise, Brain Function and MDD

The term physical exercise is often used interchangeably with physical activity, although these are not exactly the same. Indeed, physical activity refers to any body movement produced by skeletal muscles that requires energy expenditure, whereas physical exercise refers to planned, structured and repetitive body movement aimed to improve or maintain physical fitness [84]. On the other hand, aerobic exercise is considered any activity that uses large muscle groups, can be maintained continuously and is rhythmic in nature. It includes cycling, dancing, hiking, jogging/long-distance running, swimming and walking. Conversely, anaerobic exercise refers to an intense exercise of a very short duration that does not rely on oxygen for energy generation in the contracting muscles. Examples of anaerobic exercise consist of the fast twitching of muscles that occurs during sprinting, high-intensity interval training and power-lifting [85].

A growing body of evidence has demonstrated that physical exercise (and primarily aerobic exercise) can not only improve physical health and reduce disease burden, but also has a positive impact on mental health. Indeed, physical exercise may have beneficial effects on several neurological and psychiatric disorders, including MDD [86,87]. Indeed, several studies have reported the beneficial effects of physical exercise in attenuating depressive symptoms in humans and reducing the risk of developing MDD. In fact, the effect of aerobic physical exercise in improving depressive symptoms (remission and response rate) is comparable to the effect of antidepressants, but with the additional benefit of improving general health [88]. A meta-analysis carried out by Pearce et al. [86] observed the significant benefits of physical exercise in reducing the risk of depression even at levels below the public health recommendations. Accordingly, a systematic review of meta-analyses concluded that physical exercise has a beneficial effect on depressive symptoms in the general population across a wide age-range [89]. Regarding anaerobic physical exercise, although there are fewer studies investigating its beneficial effects on depression, there is also evidence that engaging in this form of physical exercise is inversely associated with depression in adults [90]. In addition, physical exercise has been shown to have the ability to positively impact global cognitive function, maintaining the integrity of hippocampal and white matter volume [91,92]. In this regard, it is interesting to note that one of the main characteristics associated with MDD is cognitive dysfunction [93]. In agreement, depressive episodes are related to reductions in brain volume, especially in the hippocampus, causing impairment in cognitive activity [94,95]. Another interesting aspect to be highlighted is that physical exercise may improve the beneficial effects of antidepressant drugs and antidepressant therapies. For example, a study that investigated the antidepressant effect of aerobic physical exercise and electroconvulsive therapy found that the combination of these strategies led to greater remission rates than each therapy alone [96]. Similarly, the effect of antidepressant drugs has also been reported to be potentiated by physical exercise [97,98].

In the last decade, the production of myokines by the skeletal muscle in response to physical exercise has been the subject of intense research. These molecules, which include irisin, insulin-like growth factor-1 (IGF-1), cathepsin B and IL-6, mediate the crosstalk between muscle and other organs, including the brain [99,100,101]. Within this scenario, physical exercise has the capability of upregulating proliferator-activated receptor gamma coactivator-1α (PGC-1α), leading to increased expression of the downstream protein fibronectin type III domain-containing 5 (FNDC5), which can be cleaved and secreted from the muscle into the bloodstream in the form of the myokine irisin [102]. Indeed, the antidepressant effect of irisin may be particularly dependent on BDNF through a PGC-1α-induced increase in neuronal expression of FNDC5 in the hippocampus [103]. Accordingly, human studies have found a positive correlation between physical exercise and increased levels of both FNDC5 and serum irisin [104,105]. It is possible that irisin increases BDNF content through activation of the cyclic adenosine monophosphate (cAMP)/PKA/CREB pathway; however, the exact receptor activated by irisin in the CNS remains to be established [106,107,108]. 

Accordingly, irisin can cross the BBB and, in the hippocampus, increase the content of BDNF, which in turn has a pro-neurogenic effect [103,109,110]. The pro-neurogenic properties of BDNF are particularly relevant in the context of depression, since impairments in neurogenesis have been correlated with behavioral deficits observed in animal models of depression [111]. Of note, this effect of physical exercise resembles the effects of antidepressants, since neurogenesis appears to mediate, at least in part, the behavioral effects of these drugs [112]. In line with the notion that an increase in brain levels of irisin may contribute to the antidepressant effect of physical exercise, central administration of irisin was shown to have an antidepressant-like effect in mice, through a mechanism that seems to involve the upregulation of genes related to neuroplasticity [113] and an increase in long-term potentiation (LTP) after electrophysiological stimulation [114]. 

Cathepsin B, another myokine produced in the muscle, can also cross the BBB and has been shown to improve memory and cognition function [115]. It is also possible that this myokine may also contribute to the antidepressant effects of physical exercise. Indeed, cathepsin B was shown to have pro-neuroplasticity effects, since treatment of hippocampal progenitor cells with recombinant cathepsin B was shown to enhance BDNF mRNA levels and the expression of doublecortin, a microtubule-associated protein that is expressed in newly generated immature neurons [116]. Accordingly, physical exercise had no effects on hippocampal neurogenesis and spatial memory function in cathepsin B-knockout mice [116]. In addition, cathepsin B levels in human plasma are positively correlated with fitness and memory [116]. Other studies have also demonstrated that physical exercise increases plasma levels of cathepsin B in middle-aged adults [117], middle-aged women [118] and open-skill athletes [119].

Of note, physical exercise has also been shown to be associated with increased levels of other growth factors besides BDNF, including IGF-1 and VEGF [120,121], both of which also have pro-neurogenic effects [122]. Therefore, the ability of physical exercise to increase neurotrophins may underlie, at least in part, its therapeutic effect in the context of MDD. In support, reduced circulating levels of neurotrophins have been directly associated with cognitive deficits and the development of MDD [123,124]. The most studied neurotrophic factor is BDNF, due to its wide-ranging role in the CNS and the periphery. Through muscle contraction during physical exercise, skeletal muscle cells and platelets may synthesize BDNF and release it into the bloodstream [99]. Yu et al. [125] demonstrated that resistance exercise positively regulates time-dependent *bdnf* mRNA and BDNF protein levels, leading to skeletal muscle regeneration. It is not well established whether BDNF from muscle cells crosses the BBB and acts in the brain to mediate muscle–brain interactions. However, it is well described that physical exercise is capable of increasing BDNF synthesis within the CNS, an effect that may be mediated by muscle-derived metabolites, particularly irisin, which can cross the BBB, as described above [99,126]. Lactate produced during moderate-to-vigorous physical exercise may also reach the brain through monocarboxylate transporters and increase BDNF expression [127]. This effect occurs through SIRT1 activity and the consequent induction of the transcriptional coactivator PGC1α and the FNDC5/irisin pathway, which in turn results in *bdnf* gene expression [31,128]. In the CNS, BDNF acts by interacting with tropomyosin receptor kinase-B (TrkB), a postsynaptic receptor present in excitatory synapses. The binding of BDNF to this receptor causes its dimerization and autophosphorylation, triggering the activation of different downstream pathways, particularly the mitogen-activated protein kinase (MAPK), phospholipase C-γ (PLCγ) and phosphatidylinositol-3-kinase (PI3K)/protein kinase B (Akt) pathways [129]. The activation of these proteins promotes several downstream pathways, including PLCγ/IP3/Ca^2+^/CaMKII/CREB, PLCγ/DAG/PKC/ERK/CREB and PLCγ/PI3K/Akt/mechanistic target of rapamycin (mTOR). The activation of these signaling pathways (Figure 2) results in pro-neurogenic and pro-survival effects, which in turn may improve psychiatric symptoms [99,100,129,130].

IGF-1 is a polypeptide hormone, secreted both centrally and peripherally, that may cross the BBB, and its release can be stimulated by a growth hormone or occur independently [131,132,133]. IGF-1 is implicated in neuronal growth, development metabolism and neuroplasticity [134], and physical exercise was shown to increase its levels in the circulation and in the CNS [135,136]. IGF-1 produced by physical exercise appears to play an important role in the upregulation of hippocampal BDNF expression given that neutralizing IGF-1 antibodies blocks the exercise-mediated increase in BDNF mRNA and protein levels [137]. Conversely, an increase in BDNF mRNA and protein levels can be seen as a result of IGF-1 binding to its receptor IGF-1R, with the consequent activation of the PI3K/Akt signaling pathway and inhibition of the enzyme GSK-3β [138]. Since the BDNF receptor TrkB appears to share downstream signaling cascades with IGF-1R in excitatory neurons, it is possible that IGF-1 also activates both the IP3/CaMKII and the Ras/ERK pathways [137,139,140]. These pathways may be responsible, at least in part, for the exercise-induced antidepressant and pro-neurogenic effects of IGF-1. In agreement, injection of IGF-1 was shown to have an antidepressant effect in rats subjected to chronic unpredictable mild stress (CUMS), through the activation of the PI3K/Akt pathway [141]. Other studies have shown that aerobic exercise and resistance training attenuated CUMS-induced depressive behavior in rats, an effect that was accompanied by an increase in IGF-1 protein content [133]. It is worth noting that in the same study, aerobic exercise promoted the expression of BDNF, mTOR, TrkB, synapsin, synaptophysin, calcium/calmodulin-dependent protein kinase II (CaMKII)β and dopamine receptor D5 (DRD5) mRNAs, while also increasing the protein content of PGC-1α, estrogen-related receptor (ERR) β and FNDC5. On the other hand, resistance training increased the levels of mTOR, Akt, synapsin and synaptophysin mRNAs, while also enhancing the protein content of IGF-1R and p-Akt [133]. 

Another growth factor that may play a role in the effects of physical exercise is VEGF, which is known as an angiogenic factor that promotes the formation of blood vessels and increases blood flow in both the periphery and the CNS [142,143]. Of note, an increase in CNS vascularization has been related to improvements in cognition [144], as well as dendritic and synaptic plasticity [145]. VEGF may also contribute to the antidepressant-like effects of physical exercise by acting in various areas of the brain (hippocampal dentate gyrus, medial prefrontal cortex and Purkinje cells of the cerebellum) [146] through the activation of VEGF receptor FLK1 [147]. Furthermore, increased levels of VEGF in the hippocampus may be associated with the activation of HCAR1 (previously known as GPR81) by lactate released during physical exercise. In turn, the activation of this receptor by lactate induces a downstream pathway, involving PI3K/Akt and extracellular-signal-regulated kinase (ERK1/2) and resulting in increased expression and secretion of VEGF [148,149].

Of special interest is the ability of lactate to elicit neurogenic and antidepressant effects, and these effects are proposed to be dependent on the conversion of lactate to pyruvate by lactate dehydrogenase with the concomitant reduction of NAD+ to NADH [71]. The increase in intracellular levels of NADH may modulate the redox state of neurons and potentiate NMDA receptor-mediated signaling [74]. Indeed, lactate was shown to stimulate the expression of synaptic-plasticity-related genes such as *Arc*, *c-fos* and *Zif268* through a mechanism involving NMDA receptor activity and its downstream signaling targets ERK1/2 [74]. Moreover, lactate also causes the addition of a lactyl group to lysine amino acid residues located in the tails of histone proteins (i.e., histone lactylation), which in turn causes the stimulation of anti-inflammatory-related genes in microglia [150]. The inhibition of the NLRP3 inflammasome complex by lactate [151] may also play a role in the antidepressant-like effect of lactate since this inflammasome has been widely recognized as being crucial for the development of depressive symptoms under inflammatory conditions [152]. The inhibition of the NLRP3 inflammasome complex by lactate can be mediated by the activation of HCAR1 (GPR81) via a mechanism involving the intracellular adaptor protein β-Arrestin 2, and by attenuating NF-kB activity, thus reducing inflammation [153,154].

The muscle also releases adiponectin during physical exercise [155,156]. Although some studies have concluded that adiponectin may not reach the brain [157,158], others have indicated that it may cross the BBB and once in the brain may exert pro-neurogenic effects in the hippocampus [156]. Of note, voluntary wheel running attenuated corticosterone-suppressed neurogenesis, enhanced plasmatic and hippocampus levels of adiponectin, and improved dendritic plasticity in the hippocampus. However, this protocol of physical exercise only reduced depression-like behaviors in wild-type mice but not in adiponectin-knockout mice [159]. Similarly, another study showed that the antidepressant and neurogenic effects of running were diminished in adiponectin-knockout mice, suggesting that adiponectin plays an important role in mediating the antidepressant and pro-neurogenic effects of physical exercise [156]. Of note, both studies suggest that the hippocampal effects of adiponectin are mediated by AMPK [156,159]. Furthermore, the adiponectin receptor agonist (AdipoRon) has also been shown to rescue cognitive function and hippocampal neurogenesis, mimicking the effects of physical exercise [160]. Other studies have shown that intracerebroventricular administration of adiponectin facilitates LTP [161]. In agreement, treatment of hippocampal slices with adiponectin resulted in increased AMPA and NMDA receptor surface expression, thus facilitating LTP [162] through enhanced NMDA-receptor function via the PI3K/Akt pathway [163]. Moreover, adiponectin is also thought to possess anti-inflammatory properties in the brain [164]. Finally, mitochondrial biogenesis can also be upregulated by adiponectin in muscle cells through activation of adiponectin receptor 1, culminating in the stimulation of calcium/calmodulin-dependent protein kinase kinase β (CaMKKβ), AMPK and SIRT1 and resulting in PGC-1α expression [165,166].

Physical exercise is also known to have a hormetic effect as it leads to the release of stress-induced factors and the consequent activation of cellular protective systems that induce redox-sensitive signaling pathways as an adaptive response to stress [167]. This mechanism is partially mediated by an increase in reactive oxygen species (ROS), which leads to the translocation of the transcription factor Nrf2 to the nucleus, resulting in the expression of antioxidant proteins [168,169]. Nuclear translocation of Nrf2 is also dependent on either PI3K/Akt induced by growth factors or an increase in the AMP/ATP ratio, which results in kinase activation (including CaMKII and AMPK) [170,171]. 

Physical exercise has also been investigated in the context of peripheral and central inflammation since it has the ability to regulate the levels of pro- and anti-inflammatory cytokines [172]. In mice submitted to a CUMS model of depression, 8 weeks of treadmill exercise significantly improves hippocampal function, reducing levels of IL-1β, NF-kB, toll-like receptor 4 (TLR-4), myeloid differentiation factor 88 (MyD88) and TNF-α and increasing levels of IL-10 and the expression of miR-223 in this structure [173]. Physical exercise also ameliorated depressive-like behaviors in ovariectomized mice, an effect that was accompanied by a reduction in the levels of IL-1β, IL-18, NLRP3, cleaved caspase-1 P10 (active caspase-1) and integrin αM (CD11b) in the hippocampus of these mice [174]. In addition, voluntary wheel running attenuated the expression of TLR-4 and Myd88 in the hippocampus of rats submitted to maternal separation [175]. Of note, regulation of pro-inflammatory cytokines in the muscle can occur due to the activation of PGC-1-α and the subsequent inhibition of the NF-KB pathway [176]. On the other hand, voluntary wheel running decreased serum levels of IL-6 and increased the levels of macrophage migration inhibitory factor (MIF) both in the serum and the hippocampus of rats subjected to forced swimming [177]. The modulation of MIF by physical exercise seems to be involved in its antidepressant effect since MIF regulates the expression of pro-neurogenic genes (including tryptophan hydroxylase 2 (*Tph2*) and *bdnf*) through the activation of MIF receptor CD74 and the downstream ERK1/2 pathway [178].

It is important to note that IL-6 can be released by muscle cells independently of NF-κB activation, indicating that IL-6 is primarily associated with metabolic processes and in this case can be considered a myokine [100]. Moreover, IL-6 can promote an increase in anti-inflammatory cytokines, such as IL-1 receptor antagonist (IL-1ra; which in turn inhibits IL-1β signal transduction) and IL-10 (which inhibits synthesis of TNF-α) [179]. The production of IL-6 in myocytes is mediated by both the Ca^2+^–NFAT (nuclear factor of activated T cells) and p38 MAPK (mitogen-activated protein kinase) pathways [180]. Furthermore, it has been reported that IL-6 regulates adult neural stem cell numbers in mice [181], and IL-6-knockout adult mice exhibited reduced neurogenesis in the hippocampal dentate gyrus and the subventricular zone [182].

Physical exercise has also an impact on the kynurenine pathway, increasing the expression of the enzyme kynurenine aminotransferase (KAT) on skeletal muscle through activation of the PGC-1α1-PPARα/δ-KAT pathway, thus promoting the peripheral conversion of KYN to KYNA, a metabolite that activates GPR35 and is unable to cross the BBB [183,184]. Reducing the peripheral levels of circulating KYN is important in preventing it from reaching the brain and decreasing the production of KYN-mediated neurotoxic and inflammatory metabolites in the brain, which could contribute to the development of depressive symptoms [184,185]. 

Several lines of evidence have indicated that physical exercise is also able to activate the endocannabinoid system [186]. This system includes the endogenous endocannabinoid araquidoniletanolamide (anandamide; AEA), which is a fatty acid that can act as an atypical neurotransmitter, released on demand postsynaptically following excitatory neurotransmission. AEA activates the cannabinoid CB1 and CB2 receptors [187]. 2-Arachidonoylglycerol (2-AG) is another well-characterized endocannabinoid that is a key regulator of neurotransmitter release. Both AEA and 2-AG can be released by neurons both centrally and peripherally in an activity-dependent manner, thus contributing to the modulation of synaptic activity and plasticity by physical exercise [188]. Moreover, aerobic exercise increased AEA levels and elevated brain hippocampal endocannabinoid CB1 receptor density in rats [189]. Further reinforcing the role of the endocannabinoid system in MDD, overexpression of CB2R was shown to decrease depressive-like behavior, whereas CB1R deficiency was associated with depressive-like behavior in rodents. In addition, in streptozotocin-diabetic rats, AEA administration improved depressive-like behavior, an effect reversed by the CB1 antagonist AM251, suggesting that the antidepressant effect of AEA is mediated, at least partially, by CB1 receptor activation. Accordingly, individuals with depression presented lower serum levels of AEA and 2-AG [190]. Moreover, a meta-analysis observed that 14 out of the 17 studies evaluated detected an increase in endocannabinoids following acute exercise [191].

## 4. Physical Exercise and Gut Microbiota

In addition to its well-recognized importance for brain function, emerging evidence has also suggested that physical exercise also has the ability to modulate gut microbiota composition, therefore improving the gut–brain crosstalk [192]. Although the exact mechanisms by which physical exercise modulates the gut microbiota are not fully understood, there is a wide range of evidence indicating that this can modulate the production of SCFAs [193]. In turn, SCFA upregulation can directly improve intestinal health, while also positively impacting brain function [40,194]. 

Of note, among SCFAs, butyrate appears to be particularly affected by physical exercise [195,196,197]. Matsumoto et al. [196] observed higher concentrations of butyrate following 5 weeks of voluntary wheel running in rats and when compared with sedentary controls. Similarly, a study by Estaki et al. [198] showed increased total SCFA production in C57BL/6 mice subjected to 6 weeks of voluntary wheel running. FMT from C57BL/6J mice exposed to exercise was also shown to increase total SCFA concentration in the feces of sedentary mice subjected to a high-fat and high-cholesterol diet when compared with animals that received FMT from sedentary animals [199]. In humans, physical exercise seems to have the same efficacy in enhancing SCFAs, since 6 weeks of physical exercise was shown to increase fecal concentrations of SCFAs in lean subjects [200]. Moreover, higher fecal levels of SCFAs have also been observed in athletes [201]. 

Butyrate is well known to inhibit histone deacetylase (HDAC) [202]. Histone acetylation is an epigenetic modification associated with the relaxation of chromatin, which in turn can enable the activation of gene transcription [203]. On the other hand, histone deacetylases are enzymes that can remove histone acetyl-groups, allowing the DNA to wrap histones more tightly, repressing transcription [202]. Butyrate has been shown to regulate gene expression by inhibiting HDACs, which can result, for example, in the modulation of the expression of tight junction proteins (e.g., zonulin, occludin and claudin) [194]. Butyrate also contributes to the strength and functioning of the intestinal barrier through the assembly of tight junctions by activating AMPK [204]. Low levels of butyrate in feces are associated with inflammatory disorders of the gut mucosa [205]. It is important to highlight that regulation of tight junction proteins in the gut is crucial for the maintenance of epithelial integrity, avoiding the translocation of pro-inflammatory endotoxins, particularly LPS, into circulation. Therefore, exercise-induced increases in gut levels of butyrate may improve the integrity of the intestinal barrier [206]. In line with this notion, trained athletes were reported to have lower circulating LPS levels when compared to sedentary individuals [207]. 

In addition to its beneficial effects on gut function, butyrate may also exert neuroprotective effects, at least partially by preventing pro-inflammatory molecules from reaching the CNS [208]. Within this scenario, it has been shown that butyrate can influence the levels of neurotrophic factors in the brain. For example, butyrate was shown to reverse the reduction in BDNF, NGF and GDNF levels in the hippocampus and frontal cortex of rats subjected to a model of bipolar disorder [62] and to improve cognitive decline, loss of neuronal spines and the reduction in BDNF induced by the NMDA receptor agonist quinolinic acid both in vitro and in vivo [209]. In agreement, butyrate has been shown to exert antidepressant-like effects in mice by leading to an increase in the levels of BDNF in the frontal cortex [66]. 

Concerning the effects of physical exercise on the composition of the gut microbiota, several studies have consistently shown that exercise can improve microbial diversity [210,211]. In animals, voluntary wheel running has been shown to positively modulate Bacteroidales S24-7 (Muribaculaceae), Clostridiaceae, Lachnospiraceae, Ruminococcaceae [195] and the Clostridiales order [196], all of which appear to be related to butyrate production. A similar study conducted by Kang et al. [212] showed reductions in Bacteroidetes and Tenericutes phyla and an increase in the Firmicutes phylum. Likewise, a study by Campbell et al. [213] observed enrichment in *Faecalibacterium prausnitzi*, *Clostridium* spp., and *Allobaculum* spp. in mice exposed to voluntary wheel running. Considering that *F. prausnitzi* is associated with butyrate production and intestinal protection, this alteration promoted by physical exercise may be beneficial to both gut and brain functions [213]. Another study showed increased levels of the Bacteroidetes phylum in the distal gut (cecum, colon) and feces [214] of rodents exposed to a treadmill exercise protocol. Of note, the increase in *Bacteroides* spp., members of the phylum Bacteroidetes, is also associated with the production of acetate and propionate [215]. Another interesting finding is that reduced levels of *A. muciniphila* are associated with depressive- and anxiety-like behaviors in animals [216,217]. Conversely, its administration in a probiotic form improves depressive- and anxiety-like behaviors [26,216,218]. Furthermore, in obese mice subjected to a high-fat diet, normalization of *A. municiphila* levels by probiotic treatment resulted in increased levels of intestinal endocannabinoids, which in turn can control inflammation, the gut barrier and gut peptide secretion [219]. In addition, *A. municiphila* is also capable of producing both propionate and acetate [220,221].

In humans, the effect of physical exercise on microbiota composition has also been reported. Women with an active lifestyle have a higher abundance of health-promoting bacterial species such as *F. prausnitzii*, *Roseburia hominis* and *A. muciniphila* when compared with sedentary women [222]. In a different study, a higher proportion of the genus *Akkermansia* was seen in male elite professional rugby athletes and subjects with low body mass index (BMI) when compared with individuals with high BMI [223]. In agreement with these findings, sedentary overweight women showed an increase in *Akkermansia* genus and a decrease in Proteobacteria phylum [224] after six weeks of an endurance exercise intervention. The Prevotellaceae family, the *Prevotella* genus and *A. muciniphila* were also positively correlated with maximal endurance and resistance exercise capacities [225]. On the other hand, reduced levels of *A. muciniphila* may be associated with depression [226] and bipolar disorder [227].

In addition to increasing the taxa of beneficial bacteria, physical exercise may also decrease the levels of some pathogenic bacteria. For example, in a study conducted in Japan with 33 elderly individuals, it was reported that a 5-week endurance exercise program reduced the abundance of *Clostridium difficile* [228], a main cause of infectious diarrhea due to the production of toxins in the host intestine [229].

Gut dysbiosis, which occurs due to a reduction in SCFA-producing bacteria, has been shown to impair muscle strength. Moreover, gut inflammation may trigger skeletal muscle atrophy due to enhanced expression of specific genes that induce muscle atrophy [230,231]. Further reinforcing the role of the gut microbiota in muscle function, a study by Cani et al. [232] reported that probiotic supplementation decreased circulating LPS and inflammation and caused a consequent increase in muscle mass in obese mice. Moreover, a relationship between age-related muscle mass wasting and fecal microbiota composition was observed in a rat model of sarcopenia, with a reduction in anti-inflammatory and pro-anabolic taxa, such as *Clostridium* XIVa cluster, *Butyricicoccus*, *Sutterella*, *Coprococcus* and *Faecalibacterium*, and genes involved in nutrient biosynthesis and catabolism [233].

The gut–muscle axis also involves the induction of AMPK phosphorylation in myotubes and skeletal muscles by SCFAs, due to their ability to increase AMP concentration and the AMP/ATP ratio [234]. Phosphorylation of AMPK results in the activation of downstream targets such as PGC-1α and p38MAPK, which in turn can promote mitochondrial biosynthesis, fatty acid oxidation and a reduction in fat accumulation in muscle [235]. Moreover, gut SCFAs can activate GPR43 in muscle cells and promote the transcription of muscle fiber genes through CaMK activation, resulting in HDAC inhibition and myocyte enhancer factor 2 (MEF2) transcriptional activation [236,237]. In addition, SCFAs also activate GPR43 and GPR41 in skeletal muscle, promoting the release of IGF-1, which activates the PI3K/Akt/mTOR pathway, thus stimulating protein synthesis [238].

Subjects with high levels of cardiorespiratory fitness seem to have an increase in Clostridiales, Roseburia, Lachnospiraceae and Erysipelotrichaceae, which appear to be able to stimulate butyrate production [198]. In obese humans, an eight-week exercise protocol was shown to increase the abundance of *Ruminococcus gauvreauii*, *Lachnospiraceae* FCS020 group and *Anaerostipes* [239]. It is worth noting that *R. gauvreauii* and *Anaerostipes* are capable of producing acetate and butyrate, respectively [240,241]. Another study evaluated the microbiota of amateur half-marathon runners and showed a significant increase in the Coriobacteriaceae and Succinivibrionaceae families [242]. Similarly, in an elegant study conducted by Scheiman and collaborators [243], an increase in the levels of the *Veillonella* genus was seen in post-race samples from marathon runners. Subsequently, a strain of *Veillonella atypica*, isolated from stool samples of these marathoners, was inoculated into CL57BL/6 mice, resulting in an increase in exhaustive treadmill run time. These results were attributed to the ability of *Veilonella* to generate propionate from lactate that enters the gut after being produced by muscle during exercise. Propionate, in turn, can serve as an energy source, thus improving exercise capacity [243]. 

It is important to note that physical exercise can increase the abundance of lactic acid bacteria, such as *Lactobacillus* and *Bifidobacterium* [211]. These bacteria may produce lactate through the fermentation of non-digestible carbohydrates, and subsequently, such lactate can be used for the production of butyrate and propionate by bacteria capable of utilizing lactate [244]. Of note, lactate can also affect the proliferation of intestinal stem cells in newborn mice and prevent injury from radiation and chemotherapy through activation of GPR81 and its downstream pathway, as well as Wnt3 signaling, which plays an important role in the maintenance of stem cell niches and their self-renewal [245]. The activation of GPR81 in the gut also regulates the inflammatory immune response through the inhibition of pro-inflammatory cytokines such as IL-6, IL-12p40, TNF-α and IL-1β and the increase in anti-inflammatory cytokine IL-10, thus maintaining gut homeostasis [246]. Notably, exercise-induced changes in the microbiota appear to be transmitted through the FMT. Sedentary C57BL/6J mice that received FMT from their exercised counterparts showed elevated GPR109A and decreased GPR43 and TNF-α in the hippocampus, as well as increased GPR109A and GPR41 expression in proximal colon tissue [199]. The same study also showed an improved insulin signaling pathway (IRS2/PI3K/Akt), increased BDNF, PSD95, and synaptophysin levels and decreased HDAC2 and HDAC3 protein expression in the hippocampus of sedentary mice that received FMT from exercised mice [199]. 

Another mechanism by which physical exercise may influence microbiota composition and, consequently, brain function is through the production of irisin. This protein was shown to reduce the degree of inflammation in a model of ulcerative colitis in mice, an effect associated with altered microbiota composition. In mice subjected to colitis, the relative abundances of Lactobacillaceae, o-Rhodospirillales and Staphylococcaceae were increased, whereas irisin treatment reverted these alterations [247]. Irisin can also prevent myocardial ischemia–reperfusion (I/R)-induced gut dysbiosis, reducing the abundance of Actinobacteriota and increasing the abundance of Firmicutes. In addition, irisin was shown to maintain intestinal barrier integrity, reduce metabolic endotoxemia through inhibition of LPS translocation from the intestine to the systemic circulation and inhibit the expression of the pro-inflammatory cytokines IL-1β, IL-6 and TNF-α [248]. Interestingly, administration of *Lactobacillus plantarum* significantly upregulated the expression of irisin in skeletal muscle in mice [249]. Another study showed increased depressive- and anxiety-like behaviors in control and CUMS Fndc5/irisin-knockout mice when compared with their wild-type controls. Furthermore, Fndc5/irisin-knockout mice also showed altered diversity and richness of their gut microbiota, with a decrease in Firmicutes and an increase in Bacteroidetes, as well as a reduction in the relative abundance of *Lactobacillus* and *Bifidobacterium* [250]. These findings establish a relationship between irisin produced by muscle during physical exercise, the gut microbiota and mood regulation. 

Physical exercise also seems to positively influence the production of the amino acids Trp, phenylalanine, and tyrosine by the gut microbiota, as seen in the feces of half-marathon runners [242] and male endurance cross-country athletes [251]. Fecal Trp levels have also been shown to be negatively correlated with *E. coli* levels and positively correlated with the *Romboutsia* genus, while also being affected by the relative abundance of *Romboutsia*, *Ruminococcocaceae* UCG-005, *Blautia*, *Ruminiclostridium* 9 and *Clostridium phoceensis* belonging to the Clostridia class [251], which are known to be able to synthesize Trp [252]. Of note, a reduction in serum Trp levels was found in subjects submitted to a cycling test [253], while decreased levels of serum Trp and increased levels of KYNA (and a consequent increase in the KYN/Trp ratio) were also reported after a single session of endurance exercise [254]. It could be hypothesized that this decrease in serum Trp is related to its uptake by the CNS, where it may be used for serotonin production [253]. Importantly, KYNA can also protect the intestinal mucosa from the formation of toxin-induced ulcers [255] and reduce inflammation triggered by LPS [256]. The activation of GPR35 receptors by KYNA may be responsible for its anti-inflammatory activity since these receptors are highly expressed in immune and epithelial cells in the gut and are associated with attenuation of the inflammatory response [184,257,258,259].

The role of extracellular vesicles (EVs) derived from the gut microbiota in the maintenance of gut immune homeostasis has also been investigated. For example, *A. muciniphila*-derived EVs ameliorated the production of the pro-inflammatory cytokine IL-6 induced by *Escherichia coli* EVs in colon epithelial cells. This study also reported that oral application of *A. muciniphila* EVs protected against dextran sulfate sodium-induced inflammatory bowel disease phenotypes [260]. More recently it has been suggested that EVs play a crucial role in the beneficial effects of physical exercise due to their miRNA content, which includes miR-21, miR-146, miR-486, miR-148a-3p, miR-223-3p, miR-142-3p and miR-191a-5p [261]. Indeed, physical exercise has been shown to induce miRNA signatures in circulating EVs related to Nrf2 signaling. In this regard, it has been suggested that miR-340-5p may enhance Nrf2 protein expression in mouse skeletal muscles following aerobic exercise [262,263].

Modulation of inflammatory states by physical exercise plays an important role in gut microbiota and health, since an increase in anti-inflammatory cytokines and anti-apoptotic proteins in gut lymphocytes and a reduction in pro-inflammatory cytokines due to physical exercise may lead to a reduction in gut inflammation [264,265,266,267]. Indeed, voluntary wheel running was shown to have an immune-regulatory impact by reducing *Turicibacter* species in feces and caecum in rodents [195]. In agreement, sedentary subjects submitted to sprint interval or moderate-intensity continuous training showed a reduction in systemic and intestinal levels of the pro-inflammatory markers TNF-α and LPS-binding protein [266]. Furthermore, obese children submitted to a 12-week strength and endurance combined training program showed a significant reduction in the Proteobacteria phylum and Gammaproteobacteria class. Moreover, this training protocol significantly reduced NLRP3 and caspase-1 protein expression in peripheral blood mononuclear cells [268]. 

The relationship between the endocannabinoid system and the gut–brain axis has also been explored. For example, *A. muciniphila* colonization enhanced intestinal endocannabinoid levels (2-AG, 2-oleoylglycerol, and 2-palmitoylglycerol) and restored gut-barrier dysfunction associated with obesity [219]. A more recent study reported that FMT from mice subjected to unpredictable chronic mild stress resulted in several abnormalities in the recipient mice, including depressive-like behavior, reduced adult neurogenesis, gut microbiota dysbiosis characterized by a decrease in *Lactobacilli* abundance, and impaired activity of the endocannabinoid system in the brain [269]. Of special relevance, microbiome-related production of endocannabinoid metabolites in the gut triggers the activation of vanilloid receptor TRPV1-expressing sensory neurons, causing an increase in dopamine levels (due to downregulation of monoamine oxidase expression) in the striatum (brain region implicated in motivation and movement initiation) during physical exercise. The stimulation of this pathway may improve physical exercise performance. Conversely, microbiota ablation with broad-spectrum antibiotics (or germ-free mice), peripheral endocannabinoid receptor inhibition, ablation of spinal afferent neurons or dopamine blockage was shown to impair physical performance [270]. As such, the maintenance of a microbiome-dependent gut–brain pathway is essential for increased levels of striatal dopamine, which is known to regulate motivation for engagement in physical exercise [270].

As discussed earlier, muscle contraction can release the myokine IL-6, which in turn, through its anti-inflammatory effects, can contribute to the maintenance of the gut microbiota and homeostasis [100]. The serum IL-6 concentration increases after physical exercise, and IL-6 can then reach the gut, where it leads to the release of GLP-1 through stimulation of intestinal L cells [271]. In turn, GLP-1 enhances insulin secretion peripherally and can enter the CNS, where it may regulate appetite and exert antidepressant effects [271,272].

The combination of physical exercise and nutritional interventions to modulate gut microbiota composition and consequently improve health and performance has provided promising results [273]. For example, the relative abundance of Anaeroplasmaceae, Christensenellaceae and *Oscillospira* in the colon and the diversity of the colon mucosa microbiota were shown to be higher in rats subjected to high-intensity interval training combined with a supplement containing polyphenol-rich extract from olive leaves, bilberry, artichoke, *Chrysanthellum* and black pepper [274]. Another study showed that a 12-week intervention program combining physical activity and n-3 polyunsaturated fatty acid (PUFA) supplementation through the addition of linseed oil to the diet of rats increased the relative abundance of *Oscillospira* and *Prevotella* in the colon microbiota [275]. In addition, a study designed to investigate the effect of supplementation of red clover (*Trifolium pratense* L., which is rich in isoflavones) on the gut microbiota found that this supplement reduced physical fatigue, possibly by enhancing the genera *Pseudobutyrivibrio*, *Parabacteroide* and *Virgibacillus* and the family Erysipelotrichaceae in female mice [276].

In humans, however, data regarding the effects of nutraceuticals on the gut microbiota composition are scarce. A recent study reported that a 4-week combined treatment of probiotics and vitamin D3 improved physical performance (extended time to exhaustion during exercise) in mixed martial arts athletes, an effect associated with an increase in Bacteroides, Peptostreptococcaceae bacterium, *Roseburia inulinivorans* species and the *Prevotella* genus and a decrease in potentially harmful Lachnospiraceae bacterium. Moreover, the Lactobacillaceae family was augmented despite the decline in bacteria from the phylum Firmicutes [277].

## 5. Conclusions and Future Directions

A dysfunction of the gut microbiota–brain axis is thought to underlie, at least in part, MDD symptoms. On the other hand, the modulation of the gut microbiota through the practice of regular physical exercise appears to be an effective strategy for maintaining mental health. The evidence reviewed in this article leads to several general conclusions regarding the possible mechanisms by which physical exercise exerts a beneficial effect against the development of depressive symptoms (Figure 3). A key event in this regard is the modulation of the gut microbiota by physical exercise, which can increase the content of SCFAs, especially butyrate, in the gut. The production of myokines, particularly irisin, also appears to be essential for gut function and to prevent depressive behavior. Both SCFAs and myokines are related to growth factor production. In this regard, the neurotrophin BDNF is known to play a key role in neurogenesis and to possess antidepressant effects. In addition, lactate produced by muscle during physical exercise, and also by the gut microbiota, may be associated with neuroplasticity and have antidepressant effects. Furthermore, physical exercise can elicit anti-inflammatory and antioxidant effects both in the periphery and centrally, and such effects can further contribute to its antidepressant effects.

Finally, physical exercise has also been shown to increase the expression of the enzyme KAT, which converts KYN into KYNA. Peripheral KYN catabolism prevents its accumulation in the brain and its consequent neurotoxic effects (due to its metabolism by microglial cells), possibly preventing the development of depressive symptoms. However, it remains to be established whether physical exercise directly impacts Trp metabolism in the gut. Other aspects that deserve further investigation include elucidating the role of physical exercise in the modulation of the endocannabinoid system and how the miRNA content of EVs relates to the gut microbiota and the development of depressive symptoms.

The limitations of our review are related to the fact that several factors may influence the diversity and composition of the gut microbiota. Indeed, the effects of exercise on the gut microbiome may be influenced by diet and stress levels, as well as the intensity, frequency and duration of physical exercise. For example, very intense and prolonged physical exercise may actually lead to deleterious effects on the gut microbiota as well as an increase in intestinal permeability [278], and these aspects are outside the scope of the present review. Another point that should be considered is that gut microbiota composition may be influenced by the method of analysis [279]. All these confounding variables may lead to results that are not comparable, and additional studies should be conducted to better characterize the effects of different protocols of physical exercise on the gut microbiota and its consequences on brain health. Future studies should also investigate the influence of supplementation with nutraceuticals on physical exercise performance, gut microbiota and mood modulation. One promising possibility is that additive or synergistic effects in physical performance and brain health may be afforded by the combination of physical exercise and antioxidant/anti-inflammatory dietary interventions.

## Figures and Tables

**Figure 1 ijms-24-16870-f001:**
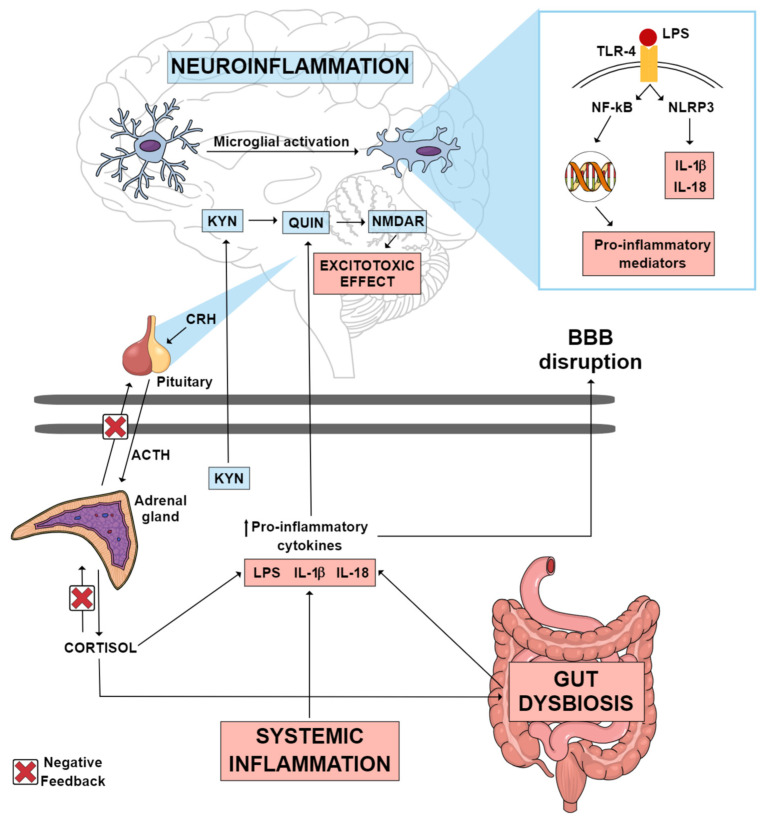
Possible inflammatory mechanisms thought to mediate neuroinflammation in MDD. Systemic inflammation, gut dysbiosis and an imbalance in the HPA axis can lead to an increase in circulating pro-inflammatory cytokines, which consequently may lead to the disruption of the blood–brain barrier. These pro-inflammatory cytokines can reach the CNS, where they can activate various mechanisms that culminate in neuroinflammation. Gut dysbiosis may be one of the factors contributing to the increase in circulating LPS. This endotoxin can reach the brain, where it may activate TLR-4 microglial receptors, promoting the transcription of pro-inflammatory mediators, as well as the activation of the NLRP3 inflammasome, which culminates in the production of IL-1 β and IL-18. A pro-inflammatory state in the CNS causes the KYN pathway to be directed towards the production of QUIN, an excitotoxic NMDAR agonist metabolite, instead of the production of KYNA, a neuroprotective NMDAR antagonist metabolite. Abbreviations: ACTH: adrenocorticotropic hormone; BBB: blood–brain barrier; CNS: central nervous system; CRH: corticotrophin-releasing hormone; HPA: hypothalamic–pituitary–adrenal; IL-18: interleukin 18; IL-1β: interleukin 1 beta; KYN: kynurenine; KYNA: kynurenic acid; LPS: lipopolysaccharide; MDD: major depressive disorder; NF-kB: nuclear factor kappa B; NLRP3: NOD-like receptor pyrin domain-containing-3; NMDAR: N-methyl-D-aspartate receptor; QUIN: quinolinic acid; TLR-4: toll-like receptor 4.

**Figure 2 ijms-24-16870-f002:**
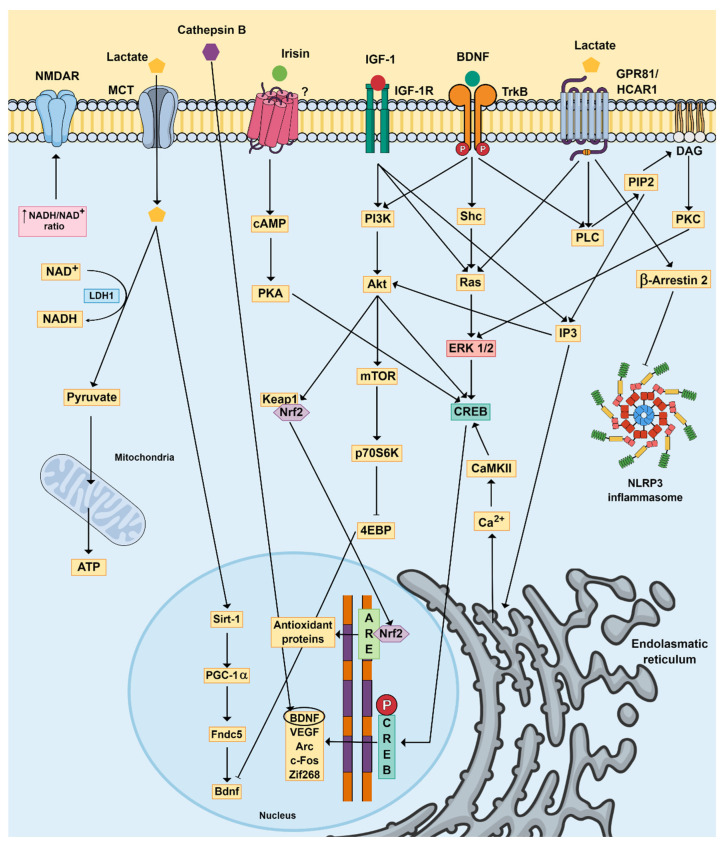
Signaling pathways through which physical exercise and skeletal muscle metabolites impact the brain. Irisin and cathepsin B are capable of increasing BDNF synthesis in the CNS. BDNF interacts with the TrkB receptor and triggers the activation of different downstream pathways, including the PLCγ/IP3/Ca^2+^/CaMKII/CREB, PLCγ/DAG/PKC/ERK/CREB and PLCγ/PI3K/Akt/mTOR pathways, leading to pro-neurogenic effects. Physical exercise can also increase other growth factors, including IGF-1 and VEGF, which further contribute to neurogenesis. Lactate can reach the brain through MCT and increase SIRT1 activity, which in turn induces the FNDC5/irisin pathway, further increasing BDNF gene expression. In addition, lactate activates the GPR81/HCAR1 receptor, which stimulates the expression of synaptic-plasticity-related genes, such as *Arc*, *c-Fos* and *Zif268*, through a mechanism involving NMDA receptor activity and its downstream signaling cascade ERK1/2. The activation of GPR81 by lactate also inhibits the NLRP3 inflammasome complex by activating β-Arrestin 2. Finally, physical exercise has a hormetic effect leading to the activation of cellular protective systems, such as the transcription factor Nrf2, resulting in the expression of antioxidant proteins. Abbreviations: 4EBP: initiation factor 4E-binding protein; Akt: protein kinase B; Arc: cytoskeleton associated protein; ARE: antioxidant response elements; ATP: adenosine triphosphate; BDNF: brain-derived neurotrophic factor; Ca^2+^: calcium; CaMKII: calcium/calmodulin-dependent protein kinase II; cAMP: cyclic adenosine monophosphate; *c-Fos*: fos proto-oncogene, AP-1 transcription factor subunit; CREB: cAMP response element-binding protein; P-CREB: phosphor-CREB; DAG: diacylglycerol; ERK 1/2: extracellular-signal-regulated kinase 1/2; Fndc5: fibronectin type III domain-containing 5; GPR81/HCAR1: G-protein-coupled receptor 81/hydroxycarboxylic acid receptor 1; IGF-1: insulin-like growth factor-1; IGF-1R: insulin-like growth factor-1 receptor; IP3: inositol trisphosphate 3; Keap1: Kelch-like ECH-associated protein 1; LDH1: lactate dehydrogenase 1; MCT: monocarboxylate transporter; mTOR: mechanistic target of rapamycin; NAD^+^: nicotinamide adenine dinucleotide; NADH: nicotinamide adenine dinucleotide reduced; Nrf2: nuclear factor erythroid-derived related factor-2; NLRP3: NOD-like receptor pyrin domain-containing-3; NMDAR: N-methyl D-aspartate receptor; p70S6K: 70-kDa ribosomal protein S6 kinase; PGC-1α: proliferator-activated receptoFndc5r gamma coactivator-1α; PI3K: phosphatidylinositol-3-kinase; PIP2: phosphatidylinositol 4,5-bisphosphate; PKA: protein kinase A; PKC: protein kinase C; PLC: phospholipase C; Shc: Src-homologous and collagen-like protein; Sirt-1: sirtuin 1; TrkB: tropomyosin receptor kinase-B; VEGF: vascular endothelial growth factor; *Zif268*: zinc-finger-containing transcription factor 268; ?: receptor not characterized.

**Figure 3 ijms-24-16870-f003:**
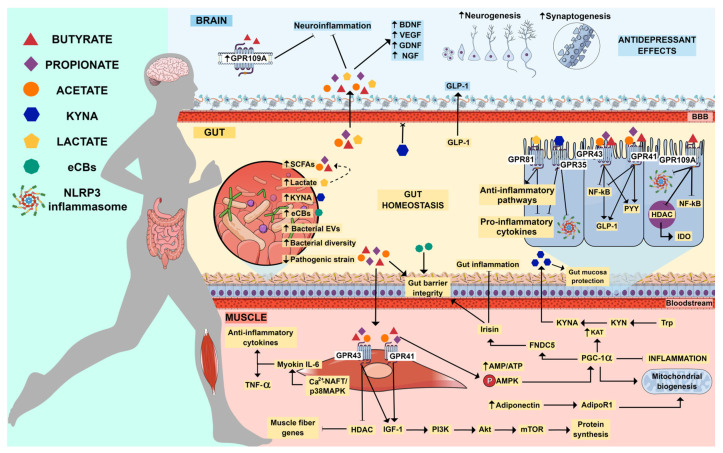
Possible mechanisms underlying the role of physical exercise in the management of MDD. Physical exercise can modulate the gut microbiota by increasing the production of SCFAs, especially butyrate. SCFAs interact with GPRs, which are involved in the production and secretion of GLP-1 and PYY, the stimulation of anti-inflammatory pathways and the inhibition of NLRP3. In particular, butyrate activates GPR109A, which not only inhibits the NLRP3 inflammasome and pro-inflammatory pathways but also inhibits HDACs that regulate gene expression, contributing to the strength and function of the intestinal barrier and to gut homeostasis. The gut microbiota also produces lactate, which can be converted into different SCFAs by several bacterial species. In the muscle, SCFAs are involved in signaling pathways that culminate in protein synthesis, mitochondrial biogenesis, the production of anti-inflammatory cytokines and the regulation of Trp metabolism, increasing the expression of KAT, which increases KYNA production. KYNA is able to protect intestinal mucosa and exerts neuroprotective effects. Modulation of physical exercise in muscle and gut metabolism can affect the brain by inhibiting neuroinflammation, enhancing the production of growth factors and stimulating neurogenesis and synaptogenesis, thus contributing to the antidepressant effects of physical exercise. Abbreviations: AdipoR1: adiponectin receptor 1; Akt: protein kinase B; AMP: adenosine monophosphate; AMPK: adenosine monophosphate-dependent kinase; ATP: adenosine triphosphate; BDNF: brain-derived neurotrophic factor; Ca^2+^: calcium; eCBs: endocannabinoids; EVs: extracellular vesicles; FNDC5: fibronectin type III domain-containing 5; GDNF: glial cell line-derived neurotrophic factor; GLP-1: glucagon-like peptide 1; GPR43/41/81;109A: G-protein-coupled receptor 41/43/81;109A; HDAC: histone deacetylase; IDO: indoleamine 2,3-dioxygenase; IGF-1: insulin-like growth factor-1; KAT: kynurenine aminotransferase; KYN: kynurenine; KYNA: kynurenic acid; mTOR: mechanistic target of rapamycin; NF-kB: nuclear factor kappa B; NGF: nerve growth factor; NLRP3: NOD-like receptor pyrin domain-containing-3; p38MAPK: p38 mitogen-activated protein kinase; pAMPK: phosphorylated AMPK; PGC-1α: proliferator-activated receptoFndc5r gamma coactivator-1α; PI3K: phosphatidylinositol-3-kinase; PYY: peptide YY; SCFAs: short-chain fatty acids; TNF-α: tumor necrosis factor α; Trp: tryptophan; VEGF: vascular endothelial growth factor.

## Data Availability

Not applicable.

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
