# Peer review of "Major Depressive Disorder and Gut Microbiota: Role of Physical Exercise"

_ijms, 2023, doi:10.3390/ijms242316870_

Round 1

Reviewer 1 Report

Comments and Suggestions for Authors

Overall, the paper is well written and is quite interesting. However, some minor suggestions should be considered by the Authors in order to improve the paper:

1. I suggest to provide a scheme describing some of the mentioned inflammatory mechanisms and their relation with the brain.

2. The Authors focused mainly on SCFAs and its possible role in MDD. However, there is no information about any direct influence of the microbiota per se. Therefore, please state some data regrading the type (composition) of bacteria in the gastrointestinal tract (ie., intestines vs. stomach). This may be of great importance as in people with different types of inflammatory bowel diseases (SIBO, IBS) the risk of MDD is higher. In fact, in some diseases of GI tract, the bacterial colonization is increased and usually includes some of the bacteria more characteristic of the colon microbiota. Therefore, it might be interesting for the reader to get to known about the specific names of bacteria which play the most important role for our mental health.

3. It would be great if the Authors could separate and divide the results describing the relationship between gut microbiota and exercises into human studies and animal.

4. Please include limitations of the study.

Comments on the Quality of English Language

the language is fine. Some minor changes should be provided

Author Response

Reviewer #1

Overall, the paper is well written and is quite interesting. However, some minor suggestions should be considered by the Authors in order to improve the paper:

  1. I suggest to provide a scheme describing some of the mentioned inflammatory mechanisms and their relation with the brain.

Author’s response: As suggested, we have provided a new figure (Figure 1), which summarizes the inflammatory mechanisms related to neuroinflammation and that are thought to be involved in the pathophysiology of Major Depressive Disorder. Please note that since, as per the IJMS guidelines, only three figures are allowed, our original Figure 1 is now the Graphical Abstract for the manuscript.

  1. The Authors focused mainly on SCFAs and its possible role in MDD. However, there is no information about any direct influence of the microbiota per se. Therefore, please state some data regarding the type (composition) of bacteria in the gastrointestinal tract (ie., intestines vs. stomach). This may be of great importance as in people with different types of inflammatory bowel diseases (SIBO, IBS) the risk of MDD is higher. In fact, in some diseases of GI tract, the bacterial colonization is increased and usually includes some of the bacteria more characteristic of the colon microbiota. Therefore, it might be interesting for the reader to get to known about the specific names of bacteria which play the most important role for our mental health.

Author’s response: We thank the reviewer for this suggestion. We have now added a paragraph to the Section “Gut Microbiota and MDD” (Section 2), which addresses the roles of beneficial bacteria in maintaining homeostasis and intestinal health, as well as their relationship with intestinal dysbiosis, and mental health, with a focus on MDD. The new paragraph reads as follows:

“Clinical evidence has shown that MDD individuals have an altered microbiota composition when compared to healthy controls, including an increased Bacteroidetes/Firmicutes ratio, which is considered a dysbiosis signature [24]. Of note, a meta-analysis study has demonstrated a decreased abundance of the bacterial families Veillonellaceae, Prevotellaceae, and Sutterellaceae, as well as the geni Coprococcus, Faecalibacterium,Ruminococcus, Bifidobacterium, and Escherichia, and an increased abundance of the family Actinomycetaceae, and the genus Paraprevotella in MDD patients compared to controls [23]. In addition, preclinical studies have shown that probiotics could improve depression-like phenotypes. For example, Tian et al. [25] reported that ingestion of the Bifidobacterium longum subspecies infantis strain CCFM687 improved stress-induced depressive-like behavior, increased BDNF levels and the abundance of butyrate-producing bacteria, and modulated the HPA axis in mice. Also, treatment with Akkermansia muciniphila improved chronic stress-induced depressive-like behavior, modulated corticosterone, dopamine, and BDNF levels, and regulated gut microbiota and metabolites in mice [26]”.

  1. It would be great if the Authors could separate and divide the results describing the relationship between gut microbiota and exercises into human studies and animal.

Author’s response: Although we have not divided the results into different sections on preclinical and clinical studies, we have carefully revised the article and tried to follow the same sequence throughout the text, first discussing preclinical studies, and then clinical ones.

  1. Please include limitations of the study.

Author’s response: As suggested, we have included the limitations of our review article in the “Conclusion and Future Directions” Section (Section 5). The new paragraph reads as follows:

“The limitations of our review are related to the fact that several factors may influence the diversity and composition of gut microbiota. Indeed, the effects of exercise on the gut microbiome may be influenced by diet, stress levels, as well as the intensity, frequency, and duration of physical exercise. For example, very intense and prolonged physical exercise may actually lead to deleterious effects on gut microbiota as well as an increase in intestinal permeability [277], and these aspects are outside the scope of the present review.  Another point that should be considered is that gut microbiota composition may be influenced by the method of analysis [278]. All these confounding variables may lead to results that are not comparable, and additional studies should be conducted to better characterize the effects of different protocols of physical exercise on gut microbiota and its consequence on brain health. Future studies should also investigate the influence of supplementation with nutraceuticals on physical exercise performance, gut microbiota, and mood modulation. One promising possibility is that additive or synergistic effects in physical performance and brain health may be afforded by the combination of physical exercise and antioxidant/anti-inflammatory dietary interventions.”

Reviewer 2 Report

Comments and Suggestions for Authors

Dear Editor,
I appreciate the opportunity to review the manuscript entitled:
"Major Depressive Disorder and gut microbiota: Role of physical exercise"

This paper focuses on key issues that are worth exploring and is well-written; however, I have some comments to help authors improve the paper:

It is not clear from the manuscript whether any effect of nutraceuticals on the microbiota has been examined. In individuals who engage in systematic athletic exercise, access to a diet supplemented with in specialty foods appears particularly available, and given the possibility of nonprescription purchase of nutraceuticals such an effect deserves to be described. I ask the authors to integrate this aspect in the text or, if not available, to integrate it in the limitations.

Author Response

Reviewer#2

This paper focuses on key issues that are worth exploring and is well-written; however, I have some comments to help authors improve the paper:

It is not clear from the manuscript whether any effect of nutraceuticals on the microbiota has been examined. In individuals who engage in systematic athletic exercise, access to a diet supplemented with in specialty foods appears particularly available, and given the possibility of nonprescription purchase of nutraceuticals such an effect deserves to be described. I ask the authors to integrate this aspect in the text or, if not available, to integrate it in the limitations.

Author’s response: We have added a paragraph on the impact of nutraceuticals on the gut microbiota in the context of physical exercise to the “Physical Exercise and Gut Microbiota” Section (Section 4). The new paragraph reads as follows:

“The combination of physical exercise and nutritional interventions to modulate gut microbiota composition and consequently, improve health and performance have provided promising results [273]. For example, the relative abundance of Anaeroplasmaceae, Christensenellaceae and Oscillospira in the colon as well as the diversity of the colon mucosa microbiota was shown to be higher in rats subjected to high-intensity interval training combined with a supplement containing polyphenol-rich extract from olive leaves, bilberry, artichoke, chrysanthellum, and black pepper [274]. Another study showed that a 12-week intervention program combining physical activity and n-3 PUFA supplementation through the addition of linseed oil to the diet of rats increased the relative abundance of Oscillospira and Prevotella in the colon microbiota [275]. In addition, a study designed to investigate the effect of supplementation of red clover (Trifolium pratense L., which is rich in isoflavones) on gut microbiota found that this supplement reduced physical fatigue, possibly by enhancing the genera of Pseudobutyrivibrio, Parabacteroide, and Virgibacillus and the family Erysipelotrichaceae in female mice.

In humans, however, data regarding the effects of nutraceuticals on gut microbiota composition are scarce. A recent study reported that a 4-week combined treatment of probiotics and vitamin D3 improved physical performance (extended time to exhaustion during exercise) in mixed martial arts athletes, an effect associated with an increase of Bacteroides, Peptostreptococcaceae bacterium, Roseburia inulinivorans species, and Prevotella genus, and a decrease of potentially harmful Lachnospiraceae bacterium. Moreover, the Lactobacillaceae family augmented despite the decline of bacteria from the phylum Firmicutes [276].”

In addition, we have also alluded to the need for further studies to investigate the effects of nutraceuticals and probiotics in “Conclusions and Future Directions” Section (Section 5) of the revised manuscript. The new paragraph reads as follows:

“Future studies should also investigate the influence of supplementation with nutraceuticals on physical exercise performance, gut microbiota, and mood modulation. One promising possibility is that additive or synergistic effects in physical performance and brain health may be afforded by the combination of physical exercise and antioxidant/anti-inflammatory dietary interventions.”